# The Relationship between Exposure to Airborne Particulate and DNA Adducts in Blood Cells in an Urban Population of Subjects with an Unhealthy Body Mass Index

**DOI:** 10.3390/ijerph19095761

**Published:** 2022-05-09

**Authors:** Alessandra Pulliero, Simona Iodice, Angela Cecilia Pesatori, Luisella Vigna, Zumama Khalid, Valentina Bollati, Alberto Izzotti

**Affiliations:** 1Department of Health Sciences, University of Genoa, 16132 Genoa, Italy; zumama.khalid@gmail.com; 2Epiget Lab, Department of Clinical Sciences and Community Health, University of Milan, 20122 Milan, Italy; simona.iodice@unimi.it (S.I.); angela.pesatori@unimi.it (A.C.P.); valentina.bollati@unimi.it (V.B.); 3Fondazione IRCCS Ca’ Granda Ospedale Maggiore Policlinico, 20122 Milan, Italy; luisella.vigna@policlinico.mi.it; 4Department of Experimental Medicine, University of Genoa, 16132 Genoa, Italy; izzotti@unige.it; 5IRCCS Ospedale Policlinico San Martino, 16132 Genoa, Italy

**Keywords:** airborne particulate, DNA adducts, blood, overweight, BMI

## Abstract

Bulky DNA adducts are a combined sign of aromatic chemical exposure, as well as an individual’s ability to metabolically activate carcinogens and repair DNA damage. The present study aims to investigate the association between PM exposure and DNA adducts in blood cells, in a population of 196 adults with an unhealthy BMI (≥25). For each subject, a DNA sample was obtained for quantification of DNA adducts by sensitive^32^P post-labelling methods. Individual PM_10_ exposure was derived from daily mean concentrations measured by single monitors in the study area and then assigned to each subject by calculating the mean of the 30 days (short-term exposure), and of the 365 (long-term exposure) preceding enrolment. Multivariable linear regression models were used to study the association between PM_10_ and DNA adducts. The majority of analysed samples had bulky DNA adducts, with an average value of 3.7 ± 1.6 (mean ± SD). Overall, the findings of the linear univariate and multiple linear regression showed an inverse association between long-term PM_10_ exposure and adduct levels; this unexpected result might be since the population consists of subjects with an unhealthy BMI, which might show an atypical reaction to airborne urban pollutants; a hermetic response which happens when small amounts of pollutants are present. Pollutants can linger for a long time in the adipose tissue of obese persons, contributing to an increase in oxidative DNA damage, inflammation, and thrombosis when exposure is sustained.

## 1. Introduction

The adverse effect of exposure to airborne particulate matter (PM) on human health is well established. The short-term effects mainly include an increase in death rates [1] and the incidence of hospital admission for cardiovascular and respiratory diseases [2]. The long-term effects include arising of chronic obstructive pulmonary diseases [3] and lung cancer, with PM being classified by IARC as a class 1 carcinogen [4]; this pathogenic relevance is mainly attributable to fine PM bearing an average diameter of less than 2.5 um, thus being able to penetrate deeply into the respiratory system [5]. PAHs and their nitro-derivatives, which are considered classic persistent organic pollutants, are noticeable components of urban air PM_2.5_. PAHs (Polycyclic aromatic hydrocarbon), such as benzo(a)pyrene and benz(a)anthracene are mutagenic and carcinogenic environmental pollutants, Ref. [6] and they could increase the lung cancer risk in the population [7]. PM_2.5_-bound nitro-polycyclic aromatic hydrocarbons (NPAHs) are mainly released from the incomplete combustion of fossil fuels and the photochemical reactions between PAHs and nitrogen dioxide. The genotoxic consequences of airborne PAH exposure have been studied extensively in molecular epidemiology research, with bulky DNA adducts being the most often utilised biomarker of exposure. High levels of such adducts have been found in the white blood cells of people who have been exposed to high levels of PAH at work [8]. Individual responses to PM are highly variable, with just a small percentage of those who are exposed experiencing negative consequences. Fragile subjects are people who are particularly vulnerable to infection, such as the elderly, children, and people who have previously been diagnosed with respiratory or cardiovascular problems [9]; however, recent epidemiological findings highlight that the number of subjects susceptible to adverse PM effects is much wider also including un-fragile subjects devoid of previous diseases and apparently healthy [10,11]. Accordingly, there is a need to develop new tools to identify subjects sensitive or resistant to PM adverse effects at the individual level, thus allowing a more personalised approach in environmental preventive medicine.

The genotoxicity of PM10 can vary spatially and temporally, which was demonstrated in nonsmoker residents of Mexico City, and that presented significantly higher levels of DNA adducts in the dry season compared to the rainy season, because PM10 concentrations decreased during the wet station [12]. Similar results have been obtained in cell models since PM10 collected in winter induces higher levels of PAH-DNA adducts in comparison to samples collected in summer in hepatocellular carcinoma cells (HepG2) [13,14]. On the other hand, PM10 from an industrial zone induced twice the formation of DNA adducts compared to PM10 obtained from a residential area of the same city [15]. These results show that measurement of DNA adduct levels can reflect the average exposure to PM and suggest that DNA damage is closely related to the intensity of air pollution and is likely with PAH concentrations [16]. 

Levels of PAH tend to be higher in PM of smaller aerodynamic diameter (PM2.5); however, studies show that PM with a larger diameter (PM10) also induce the formation of adducts. Lung epithelial A549 and BEAS-2B cells showed DNA PAH-Adducts after exposure to PM10 and PM2.5 associated with PAH bioactivation, which is reflected in increased expression of the CYP1A1 and CYP1B1 gene and protein activity after 24 h of exposure to PM10 and 24, 48 and 72 h of exposure to PM2.5 [17,18].

Higher particulate matter with aerodynamic diameter ≤ 1 μm exposure was associated with adverse changes in blood lipid levels and dyslipidemias. Each 1 μg/m^3^ increase in particulate matter with aerodynamic diameter ≤ 1 μm was associated with 6% higher risks of hypercholesterolemia [11]. Inhaled particles cause oxidative stress and inflammatory responses in the lungs, which result in the release of molecular signals into the circulatory system [19]. Numerous studies have reported evidence of the association between environmental stressors and human health. For example, environmental cues such as nutrient intake can interact with DNA structures and alter transcriptional profiles, which could elicit stable changes in the ageing of the organism. Exposure to modest doses of environmental chemicals causes an environmentally induced change in the phenotypic, resulting in a better adaptive response to the subsequent higher dose [20]. PM collected from urban air can induce the formation of multiple DNA adducts in the lungs as identified by sensitive^32^P post-labelling methods [21]. Nitropyrene derivatives are the main contributors to PM genotoxicity, as demonstrated by comparing nuclease P1 and butanol enrichment in 32P post-labelling analysis [22,23]. The current literature supports the positive association between particulate levels and adduct quantity in populations with a normal BMI [16,17,24]; therefore, the aim of this study was to study subjects with high BMI. The present study aims to investigate the association between PM exposure and DNA adducts in blood cells, in a population of adults with an unhealthy BMI (≥25).

## 2. Materials and Methods

### 2.1. Subjects

The study population includes 196 obese subjects selected among participants of the cross-sectional Susceptibility to Particle Health Effects, miRNAs and Exosomes (SPHERE) study [25].

The SPHERE study population has been previously described in detail. Briefly, subjects were recruited from the Center for Obesity and Work (Department of Preventive Medicine, IRCCS Ca’ Granda—Ospedale Maggiore Policlinico) in Milan. Study participants reside in the Lombardy Region which is situated in the Northern part of Italy, and Milan is the regional capital. Subjects were recruited according to the following eligibility criteria: (a) older than 18 years at enrolment; (b) overweight/obese according to body mass index (BMI): overweight, BMI between 25–30 kg/m^2^; obese: BMI of 30 kg/m^2^ or more; (c) resident in the Lombardy Region at the time of recruitment. Exclusion criteria were: previous diagnosis of cancer, heart diseases, stroke, other chronic diseases, or known diagnosis of diabetes. A lifestyle questionnaire was fulfilled by each study participant collecting information on socio-demographic data. Each participant provided written informed consent, which was approved by the Ethics Committee of Fondazione IRCCS Cà Granda Ospedale Maggiore Policlinico (approval number 1425).

### 2.2. Exposure Assessment

We obtained recordings of daily air levels of PM10 from the Regional Environmental Protection Agency (ARPA Lombardy) measured from monitoring stations throughout the Lombardy region, in Italy, on the day of recruitment and for 365 days prior. We assigned to each participant the daily mean PM10 concentration of the monitor nearest to the Center for Obesity and Work on the day of recruitment. Individual PM10 exposure was then assigned to each subject by calculating the mean of the 30 days (short-term exposure), and of the 365 (long-term exposure) preceding enrolment [25]. In the case of incomplete series, each missing value was imputed by using an algorithm that integrates the annual average of the incomplete series and PM10 concentrations of the nearest and more correlated monitors.

### 2.3. Sample Collection. Blood Analyses. DNA Extraction

Seven millilitres of whole blood were collected into EDTA tubes from each participant by venous phlebotomy. After centrifuging the blood tubes at 1200 *g* for 15 min to separate plasma, buffy coat, and erythrocytes. Genomic DNA was extracted from the buffy coat fraction (including lymphocytes, monocytes, granulocytes, and platelets) using the Wizard Genomic DNA Purification Kit (Promega; Madison, WI, USA) according to the manufacturer’s instructions and used to perform DNA adduct quantification. The concentration of the purified DNA was measured using the NanoDrop-1000 spectrophotometer (Thermo Fisher Scientific; Waltham, MA, USA).

### 2.4. DNA Adducts Analysis by ^32^P Post-Labelling

Bulky lipophilic DNA adducts were enriched by nuclease P1 digestion and detected by 32P-post-labeling, as described previously [26]. Autoradiography was performed by using a ^32^P Instant Imager Electronic Autoradiographic System equipped with Instant Quant software (model A2024; Packard, Meriden, CT, USA). The relative adduct labelling index (cpm (counts per minute) adduct/cpm normal nucleotides) was calculated. DNA adducts, detected by 32P-imaging (Instant Imager, Packard, Meriden, CT, USA), were quantified by calculating the ratio between cpm detected in DNA adducts and cpm in normal nucleotides and expressed as adducts/10^8^ nucleotides.

### 2.5. Statistical Analyses

Standard descriptive statistics were performed on all variables. Continuous variables were expressed as the mean ± standard deviation (SD) or as the median with interquartile range (Q1–Q3), as appropriate. Categorical data were reported as frequencies with percentages. We evaluate by univariate linear regression models the association with DNA adducts of a series of covariates potentially associated with the outcome. These variables were: patient characteristics (sex, BMI, age, smoking habits, and diabetes), hematological parameters (protein C reactive and hemochrome), season, and apparent temperature. Variables associated with univariate analysis with DNA adduct were included in a first multivariate model (model 1) together with a selection of variables included a priori irrespective of their statistical significance (diabetes, smoking habits, age, BMI). We subsequently performed a second model excluding variables not significant in multivariate analysis. The best model selection (model 2) was based on the minimization of the Akaike information criterion and maximization of the explained variance of the model. The final models were adjusted for smoking habits, season, apparent temperature, white blood cells, and platelets. Estimated effects are reported as β and 95% confidence intervals (CI) associated with an increase of 1 unit in each independent variable. All statistical analyses were performed by using SAS 9.4 statistical software (SAS Institute Inc., Cary, NC, USA).

## 3. Results

A total of 196 subjects were involved in the present study. The main characteristics of the study subjects are shown in Table 1. 

All subjects were Caucasians, residents in North Italy, aged between 37 and 64 years; 25% were males. The majority of study participants were never smokers (53%). Only 42 subjects (21.4%) were overweight (BMI less than 29.9 kg/m^2^, with a minimum BMI equal to 25 kg/m^2^); 80 (40.8%) were obese (BMI 30.0–34.9 kg/m^2^) and 74 (37.8%) were severely obese, whilst 25 subjects had diabetes. The median values of short- and long-term exposure to PM are reported herein.

Table 2 shows high levels of PM exposure ranging from 23.1 to 79 µg/m^3^ when considering the month before enrollment and 43 and 49.1 in the year before enrollment.

Table 2 reports the PM10 level averaged over 30 days (short term) and over 365 days (long term) recorded from the three monitoring stations in Milan in the Lombardy region, which is located in the Po Valley, known to be a region with high pollution levels in Italy. The EU Ambient Air Quality Directives stated a limit value of 40 µg/m^3^ yearly and a daily mean of 50 µg/m^3^ not to be exceeded on more than 35 days per year. The concentrations of particular matters in this region remain in the atmosphere for long periods and maintain much higher levels, both in terms of time and quantity, than the limits set. The Italian Po Valley situation is particularly serious, and the concentrations exceeded the limits for over 50 days, with daily average values close to 50 µg/m^3^. [27]. The mean we calculated derives from three monitors whose maximum values reach extremely high values, as reported in the table below, reporting the description of the entire series of the 3 monitoring stations of Milan during the study period; moreover, although the annual averages might have a limited variance, a great source of variability derives also from the subject-specific day of enrolment (span of 5 years).

Bulky DNA adducts were detected in the majority of tested samples with an average value of 3.7 ± 1.6 (mean ± SD). The interindividual variation was remarkably high accounting for 51.6 ± 17.0 (mean ± SD). The number of subjects having DNA adducts below the sensitivity of the used methods was minimal (*n* = 10 out of 196 subjects, i.e., 5%). At autoradiography, 8 samples (4%) revealed the presence of a diagonal radioactive zone, indicating exposure to high doses of multiple adducts. Autoradiograms revealed the presence of multiple adducts in the vast majority of examined samples. Examples of autoradiography patterns obtained are reported in Figure 1.

These findings reflect the exposure to a mixture of multiple genotoxic agents, as typically occurs for airborne pollutants in urban areas.

Univariate linear regression analyses (Table 3) showed a positive association between DNA adducts measured in peripheral blood leukocytes, season (in particular summer) and apparent temperature, whereas a negative association was found with white blood cells and platelet count.

Results of univariate and multivariate analyses of the association between short-term PM exposure are reported in Table 4. Only in the univariate analysis, an increase of 1 µg/m^3^ in PM was associated with a significant reduction in DNA adducts; however, when adjusting for other covariates the association disappeared.

When we evaluated long-term PM exposure, a negative statistically significant association was found in both the adjusted model 1 and model 2 showing a reduction of about 0.50 for every increase of 1 µg/m^3^ in PM (Table 5).

## 4. Discussion

In metropolitan areas, air pollution and particulate matter have been linked to higher mortality from cardiovascular and respiratory disorders, as well as an increased risk of cancer. It is thought that smaller PM, particularly nanoparticles, penetrate more into the deeper layers of the skin. Furthermore, PM can absorb PAH on their surface in highly polluted areas, which would turn them into the major culprit for generating reactive oxygen species [28]. Recently, a retrospective study indicated that the nonlinear cycling pattern of the aromatic amines correlated with the PM_10_ concentration [29]. Particulate matter’s carcinogenic effects have been linked to the presence of certain chemicals, such as polycyclic aromatic hydrocarbons [30]. The latter may produce bulky DNA adducts, which could be used as possible cancer risk markers [30]. We studied whether the association between airborne particle exposure and DNA adducts in blood cells in an urban population is regulated by adaptation events and individual characteristics in a group of (*n* = 196) people living in northern Italy.

Our findings show an inverse association between PM exposure (particularly long-term exposure) and DNA adducts; this association differs from what is seen in the general population as the findings show that the PM effect in overweight people is specific. Indeed, in both experimental animals and the general population, a linear dose-response association exists between the amount of airborne particulate and the formation of DNA adducts. A possible explanation is related to the peculiar characteristic of the study subjects which include only overweight and obese subjects.

The bulky lipophilic DNA adducts discovered by 32P post labelling following butanol enrichment, the approach used in the present study, are mostly bulky lipophilic DNA adducts [30,31]. Lipophilic genotoxic pollutants tend to be sequestered into adipose tissue, as is well demonstrated for chlorophenols [13,32]. Accordingly, it is conceivable that the inverse relationship observed is due to the trapping of lipophilic polycyclic hydrocarbon metabolites in adipose tissue, where these pollutants accumulate and may persist for a long time in the organism. Their mobilization mainly occurs during adipose tissue exceedingly fast decrease, as typically occurs in medical uncontrolled slimming diets that, because of this situation, represent a risk factor for health [33]. It is well established that obesity causes systemic inflammation [34].

Body fat amount is a major factor affecting the toxicodynamics of environmental pollutants. Trapping of lipophilic pollutants inside fat tissue does not represent detoxification; indeed, these genotoxicants may be released again from adipose tissue into the bloodstream, especially when a decrease in fat tissue occurs. These toxicodynamic mechanisms are well known for lipophilic carcinogens such as chlorophenols [17,35]; however, to date, no data have been collected dealing with PM.

Exposure of our subjects was to a complex mixture of airborne pollutants, including a small contribution from B(a)P; this is the reason why the 32P post-labelling method was used for DNA adduct analysis. Indeed, this is the only method able to (a) evaluate all together multiple adducts formed by exposure to a mixture; (b) detect adducts of unknown molecular structures.

This trapping situation is one of the multiple mechanisms activated by exposed organisms to attenuate the adverse effects of exposure to environmental pollutants, a situation referred to as a ‘hermetic response’ [27,36]. A hermetic response causing the occurrence of a negative relationship between the level of environmental exposure and the number of DNA adducts in blood cells has been previously reported in populations in the Czech Republic [15,37,38,39]. The fact that in this study the inverse relationship is more remarkable for long-term exposure (1-year average) than for short-term exposure (1-month exposure) supports this interpretation of our findings.

A possible limitation of the present study is that we used PM_10_ instead of PM_2.5_ to estimate individual exposures because the PM_10_ dataset was more complete and characterised by a better spatial resolution in the study area; however, in the Lombardy Region, PM_10_ is mainly constituted by fine particles, and PM_2.5_ represents 58–94% of PM_10_ [11]. In addition, unfortunately, personal exposure monitoring data are not available due to the large study sample; thus, we could not account for indoor air pollution [40].

## 5. Conclusions

In conclusion, our data suggest that overweight people react to airborne urban contaminants in an unusual way. When exposed to modest quantities of contaminants, the hermetic response occurs. Pollutants, on the other hand, can stay in the adipose tissue of overweight people for a long time, contributing to an increase in oxidative DNA damage, inflammation, and thrombosis when exposure is prolonged. The relationship between non-communicable disease risk and DNA adduct formation has been already explored and established in a healthy population. The results presented herein indicate that obesity is a confounding factor able to influence the relationship between DNA adduct formation and exposure to environmental pollutants. The unexpected inverse association between long-term PM10 exposure and adduct levels may be related to (a) trapping of lipophilic genotoxic into adipose tissue [34,35]; (b) adaptive mechanisms triggered by low-level environmental exposure. A high body mass index makes what would be a low exposure for a person with a normal or low BMI, higher, as for example occurring in children.

## Figures and Tables

**Figure 1 ijerph-19-05761-f001:**
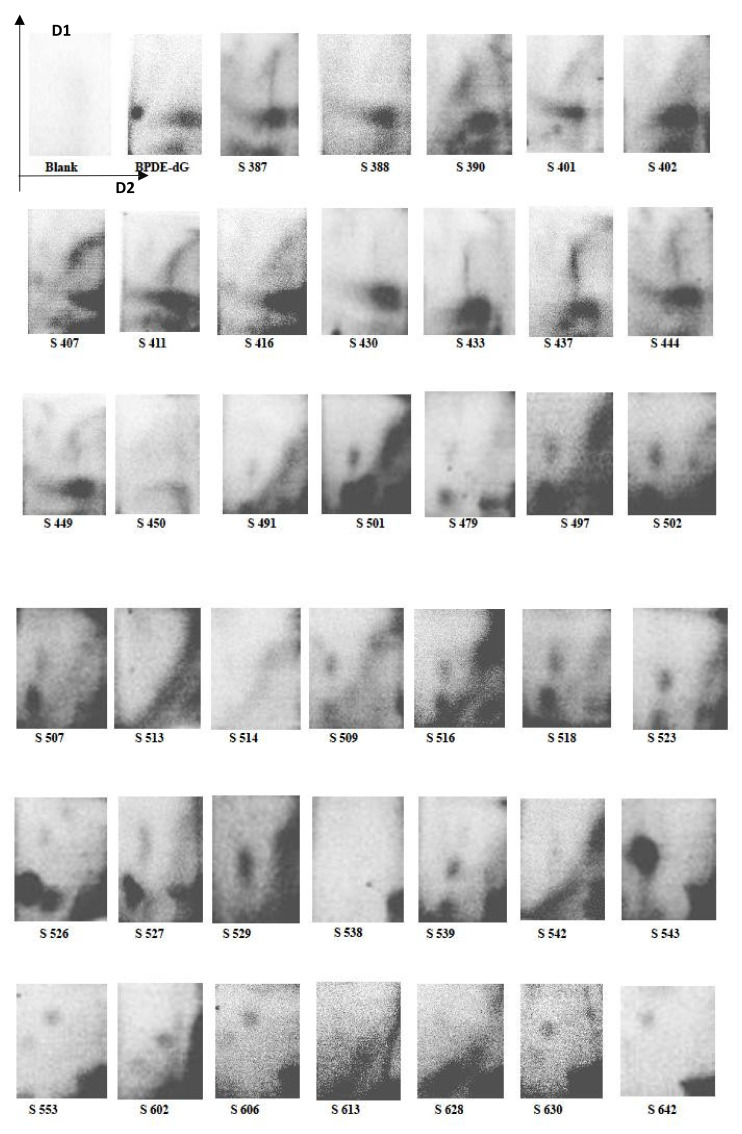
Examples of autoradiographic patterns of DNA adducts as detected by 32P post-labelling in thin layer chromatography sheets. The origin of the multidirectional chromatography was located in the left bottom corner of the sheet. Blank, DNA free negative control samples; BPDE-dG, reference positive standard; S387–S642 samples codes. A variety of DNA adduct patterns were detected including barely detectable single spots (e.g., S642), multiple spots either weak (e.g., S553) or strong (e.g., S526), diagonal radioactive zone (e.g., S516), multiple spots and diagonal radioactive zone (e.g., S516), negative samples (e.g., S538).

**Table 1 ijerph-19-05761-t001:** Characteristics of study participants.

Characteristics	
Study participants, *n*	196
Sex, *n* (%)	
* Male*	49 (25.0%)
* Female*	147 (75.0%)
Age, years, mean ± SD,	50.6 ± 13.3
Season of enrolment	
* Winter*	54 (27.6%)
* Spring*	18 (9.2%)
* Summer*	36 (18.4%)
* Autumn*	88 (44.9%)
BMI, mean ± SD, Kg/m^2^	34.2 ± 5.3
BMI classification, Kg/m^2^	
* 25–29.9 (overweight)*	42 (21.4%)
* 30–34.9 (obese)*	80 (40.8%)
* 35–39.9 (severely obese)*	74 (37.8%)
Smoking, *n* (%)	
* Never smoker*	104 (53.1%)
* Ex-smoker*	65 (33.1%)
* Actual smoker*	27 (13.8%)
Diabetes, *n*%	
* Diabetics*	25 (12.8%)
* Prediabetics*	87 (44.4%)
* Normal*	79 (40.3%)
* Missing*	5 (2.5%)
CRP, mg/dL, mean ± SD	0.5 ± 0.7
Hemochrome, 10^3^ cell/µL, mean ± SD	
* White blood cells*	6.8 ± 1.5
* Red blood cells*	4.8 ± 0.4
* Hemoglobin*	13.7 ± 1.4
* Hematocrit*	40.5 ± 3.4
* Mean Corpuscular Volume*	84.6 ± 6.9
* Platelets*	258.5 ± 64
* Neutrophils, %*	58.3 ± 7.2
* Eosinophils, %*	2.3 ± 1.3
* Lymphocytes, %*	31.3 ± 6.6
* Monocytes, %*	7.6 ± 2
* Basophils, %*	0.5 ± 0.3
* Granulocytes, %*	61.1 ± 6.9

**Table 2 ijerph-19-05761-t002:** Description of exposure to PM_10_.

	Median	IQ Range	Minimum	Maximum
PM_10_ short term, µg/m^3^	52.7	34.3	23.1	79
PM_10_ long term, µg/m^3^	44.7	3.4	43	49.1

**Table 3 ijerph-19-05761-t003:** Association between potential covariates and DNA adducts in univariate linear regression analysis. Statistically significant results are highlighted in bold character in *p*-Value column.

Variable	Estimate	95% LCI	95% UCI	*p*-Value
Season				**(0.001)**
* Spring*	0.122	−0.688	0.933	0.767
* Summer*	1.281	0.640	1.921	**<0.001**
* Autumn*	0.520	0.005	1.035	0.048
* Winter*	Reference			
Apparent temperature, celsius	0.023	0.003	0.044	**0.024**
Smoking habits				(0.211)
* Ex-smoker*	0.186	−0.299	0.672	0.450
* Actual smoker*	−0.445	−1.108	0.219	0.188
* Never smoker*	Reference			
Diabetes				(0.865)
* Diabetics*	0.164	−0.546	0.873	0.650
* Prediabetics*	0.105	−0.375	0.586	0.666
* Normal*	Reference			
Sex				
* Male*	−0.070	−0.579	0.439	0.787
* Female*	Reference			
Age, years	0.005	−0.011	0.022	0.515
BMI, kg/m^2^	−0.013	−0.054	0.029	0.544
CRP, mg/dL	−0.141	−0.477	0.195	0.408
Hemochrome, 10^3^ cell/µL				
* White blood cells*	−0.219	−0.366	−0.072	**0.004**
* Red blood cells*	−0.142	−0.666	0.382	0.594
* Hemoglobin*	0.002	−0.157	0.161	0.983
* Hematocrit*	−0.006	−0.071	0.060	0.862
* Mean Corpuscular Volume*	0.005	−0.027	0.038	0.752
* Platelets*	−0.004	−0.007	−0.001	**0.021**
* Neutrophils, %*	−0.018	−0.049	0.013	0.263
* Eosinophils, %*	−0.026	−0.204	0.151	0.771
* Lymphocytes, %*	0.019	−0.014	0.053	0.257
* Monocytes, %*	0.031	−0.079	0.141	0.577
* Basophils, %*	0.411	−0.369	1.190	0.300
* Granulocytes, %*	−0.019	−0.052	0.013	0.238

**Table 4 ijerph-19-05761-t004:** Association between PM_10_ short term and DNA adducts from univariate and multivariate regression analysis. Statistically significant results are highlighted in bold character in *p*-Value column.

	Regression Coefficient	95% LCI	95% UCI	*p*-Value
**Univariate model**				
PM_10_ short term	−0.020	−0.033	−0.008	**0.002**
**Multivariate models**				
**Model 1 ***				
PM_10_ short term	−0.023	−0.070	0.024	0.327
BMI	−0.010	−0.052	0.033	0.655
Sex				
* Male*	−0.341	−0.876	0.194	0.210
* Female*	Reference			
Diabetes				(0.468)
* Diabetics*	*0.289*	*−0.429*	*1.006*	*0.428*
* Prediabetics*	*0.288*	*−0.191*	*0.767*	*0.237*
* Normal*	Reference			
Smoking habits				0.119
* Ex-smoker*	0.230	−0.251	0.711	0.346
* Actual smoker*	−0.509	−1.171	0.153	0.131
* Never smoker*	Reference			
Season				**(0.036)**
* Spring*	0.176	−1.487	1.840	0.835
* Summer*	1.403	−0.261	3.067	0.098
* Autumn*	0.490	−0.387	1.367	0.272
* Winter*	Reference			
White blood cells	−0.065	−0.226	0.095	0.423
Platelets	−0.004	−0.008	−0.001	**0.027**
Apparent temperature	−0.042	−0.098	0.014	0.137
**Model 2 ***				
PM_10_ short term	−0.038	−0.082	0.007	0.095
Smoking habits				0.062
* Ex-smoker*	0.243	−0.227	0.711	0.309
* Actual smoker*	−0.573	−1.210	0.064	0.078
* Never smoker*	Reference			
Season				0.062
* Spring*	−0.252	−1.839	1.335	0.754
* Summer*	1.038	−0.548	2.624	0.198
* Autumn*	0.281	−0.551	1.113	0.506
* Winter*	Reference			
White blood cells				
Platelets	−0.004	−0.007	−0.001	**0.021**
Apparent temperature	−0.052	−0.107	0.003	0.066

* Model 1 is adjusted for covariates associated in univariate analysis with DNA adduct and variables included a priori (diabetes, smoking habits, age, and BMI). Model 2 excluded variables not significant in multivariate analysis and was the best model selection based on the minimization of the Akaike information criterion and maximization of the explained variance of the model. For categorical variables, we reported in brackets the overall *p*-value.

**Table 5 ijerph-19-05761-t005:** Association between PM_10_ long term and DNA adducts from univariate and multivariate regression analysis. Statistically significant results are highlighted in bold character in *p*-Value column.

	Regression Coefficient	95% LCI	95% UCI	*p*-Value
**Univariate analysis**				
PM_10_ long term	−0.22	−0.34	−0.11	<0.001
**Multivariate analysis**				
**Model 1** *****				
PM_10_ long term	−0.52	−0.85	−0.20	**0.002**
BMI	−0.01	−0.05	0.03	0.741
Sex				
* Male*	−0.46	−0.98	0.07	0.089
* Female*	Reference			
Diabetes				(0.595)
* Diabetes*	0.22	−0.48	0.92	0.534
* Prediabetes*	0.24	−0.23	0.70	0.322
* Normal*	Reference			
Smoking habits				0.086
* Ex-smoker*	0.26	−0.21	0.73	0.271
* Actual smoker*	−0.51	−1.16	0.13	0.120
* Never smoker*	Reference			
Season				**0.009**
* Spring*	−1.08	−2.74	0.58	0.202
* Summer*	0.37	−1.22	1.97	0.644
* Autumn*	−0.38	−1.36	0.60	0.449
* Winter*	Reference			
White blood cells	−0.06	−0.22	0.09	0.442
Platelets	0.00	−0.01	0.00	**0.021**
Apparent temperature	−0.06	−0.10	−0.01	**0.018**
**Model 2** *****				
PM10 long term	−0.56	−0.86	−0.25	**<0.001**
Smoking habits				(0.055)
* Ex-smoker*	0.23	−0.23	0.68	0.328
* Actual smoker*	−0.58	−1.20	0.04	0.066
* Never smoker*	Reference			
Season				**(0.004)**
* Spring*	−1.22	−2.78	0.34	0.124
* Summer*	0.28	−1.23	1.79	0.717
* Autumn*	−0.45	−1.35	0.46	0.335
* Winter*	Reference			
White blood cells	0.00	−0.01	0.00	**0.025**
Apparent temperature	−0.06	−0.10	−0.01	**0.014**

* Model 1 is adjusted for covariates associated in univariate analysis with DNA adduct and variables included a priori (diabetes, smoking habits, age, and BMI). Model 2 excluded variables not significant in multivariate analysis and was the best model selection based on the minimization of the Akaike information criterion and maximization of the explained variance of the model. For categorical variables, we reported in brackets the overall *p*-value.

## Data Availability

The datasets used and/or analysed during the current study are available from the corresponding author on reasonable request.

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
