# Peer review of "The Relationship between Exposure to Airborne Particulate and DNA Adducts in Blood Cells in an Urban Population of Subjects with an Unhealthy Body Mass Index"

_ijerph, 2022, doi:10.3390/ijerph19095761_

Round 1

Reviewer 1 Report

This article focuses on current and important issues highlighting the relationship between air pollution and the incidence of cardiovascular and lung diseases. Numerous studies have shown that air pollution causes a wide variety of health consequences. These can be more or less life-threatening. This article attempts to examine the relationship between the presence of polluted air, i.e. gaseous as well as particulate pollution, and the incidence of various diseases and the accumulation of harmful compounds in adipose tissue. I agree with the authors that one of the more troublesome human and environmental manifestations of ambient air pollution is the accumulation of pollutants in the ground layer. The article has an adequate theoretical basis, relevant information and analysis, good partial (in the article) and final (in the conclusion) conclusions. The article uses original research by the author and cited research by other researchers, which enriches its content. It is written in good language and is based on the analysis of current and well-chosen literature, although it could be further enriched with other items, e.g. P.O. Czechowski, A. Badyda, E. Czermanski, A. Oniszczuk-JastrzÄ…bek.  The research models were applied correctly. Systematics of models is not a simple issue, as the differentiation of model types results mainly from their precisely defined purpose. Therefore the article should be treated as an interesting introduction to a very important issue and treated as a scientific article.

Author Response

Dear Editors,

We would like to thank you for considering the manuscript entitled “The relationship between exposure to airborne particulate and DNA adducts in blood cells in an urban population of subjects with an unhealthy body mass index

” by A. Pulliero et al., and for sharing the Reviewers’ comments that certainly helped in improving the quality of the manuscript (IJERPH 1668622 ). We appreciated the Reviewers’ comments, and we revised the manuscript accordingly. Please find enclosed to the submission of the revised version of the manuscript the point-by point reply to the Reviewers’ comments. For clarity’s sake, changes in the revised MS are wrote in red colour.

We hope that the revised version of our MS will be now suitable for publication in the IJERPH.

Accordingly, we prepared a revised version of the manuscript acknowledging Referees’ and Editor’s comments as below specified:

Reviewer 1:

COMMENT 1. The article has an adequate theoretical basis, relevant information and analysis, good partial (in the article) and final (in the conclusion) conclusions. The article uses original research by the author and cited research by other researchers, which enriches its content. It is written in good language and is based on the analysis of current and well-chosen literature, although it could be further enriched with other items, e.g. P.O. Czechowski, A. Badyda, E. Czermanski, A. Oniszczuk-JastrzÄ…bek.  The research models were applied correctly. Systematics of models is not a simple issue, as the differentiation of model types results mainly from their precisely defined purpose. Therefore the article should be treated as an interesting introduction to a very important issue and treated as a scientific article.

ANSWER 1 We wish to thank the Reviewer for the kind and positive comments. We have revised the

manuscript, taking into account the Reviewer’s concerns below.

Reviewer 2 Report

The paper of  Pulliero et al. “The relationship between exposure to airborne particulate and 2 DNA adducts in blood cells in an urban population of subjects 3 with an unhealthy body mass index “ analyzed the possible correlation between air pollution exposure and DNA adducts

The authors conclude that… Overall, the 25 findings of the linear univariate and multiple linear regression showed an inverse association between long-term PM10 exposure and adduct levels.  The unexpected results, in my opinion, maybe due to an incorrect experimental model.

  1. Why These people also have altered values of different markers (inflammatory for example) which can alter the results analyze the obese people?
  2. First of all, I would analyze this possible correlation in people of normal weight
  3. In part 2.2. Exposure assessment is not clear how exposure is determined
  4. Where do these people live? What is their daily activity? How long they stay away from home and at home (indoor pollution) etc; this impossibility of determining exposure makes it impossible to say whether or not there is a correlation between air pollution and DNA adducts.
  5. If the calculated average values (table 2) are also correct, the exposure (except PM 10 short term, μg / m3 maximum value) can be said to be equal to or very close to the normally required value by 2008/50/CE PM10 50 µg/m3/day.

It could be a low value to see some variation; if at this value were found significant adduct DNA adducts, it would be very worrying!

Therefore, it is necessary to calculate the exposure much more precisely, treat the summer exposure differently from the winter one and divide the population into exposure groups to have reliable results.

  1. Figure 1 is impossible (even when zooming in on the screen) to see and understand (the methods section does not adequately describe this analysis). Is it an enlargement? What is the direction of chromatography?
  2. Line 182-184, the authors found a positive association with air pollution exposure and DNA adducts .. showed a positive association between DNA adducts, season (in particular summer) …… in which cells?.... and apparent temperature, whereas a negative association was found with white blood cells and platelets….?
  3. Winter PM is known to be the richest in PAH (Camatini, et al. 2010. PM10-biogenic fraction drives the seasonal variation of pro-inflammatory response in A549 cells. Environ. Toxicol., doi:10.1002/tox.20611; Perrone, et al., 2010. Seasonal variations in chemical composition and in vitro biological effects of fine PM from Milan. Chemosphere 78, 1368–1377) Consequently, if the PAHs induce DNA adducts, why is there no positive correlation between PM winter and DNA adducts, but there is with PM summer?

The subject matter is exciting, but it is impossible to reach a conclusion with this model.

Author Response

Reviewer 2:

COMMENT 1 The authors conclude that… Overall, the 25 findings of the linear univariate and multiple linear regression showed an inverse association between long-term PM10 exposure and adduct levels.  The unexpected results, in my opinion, maybe due to an incorrect experimental model.

ANSWER 1. The Reviewer makes a valid point. The results could lead to misinterpretation or even doubts about the performance of laboratory assays. Follow-up studies, however, confirmed these results, and due to the application of molecular these results were followed by other studies suggesting no increase, or even lower levels of DNA damage in subjects residing in the highly polluted Moravian-Silesian Region. It could suggest possible interpretations and a new view on the data obtained from biomonitoring studies. (P. Rossner Jr., V. Svecova, J. Schmuczerova, A. Milcova, N. Tabashidze, J. Topinka, A. Pastorkova, R.J. Sram, Analysis of biomarkers in a Czech population exposed to heavy air pollution. Part I: bulky DNA adducts, Mutagenesis 28 (2013) 89–95.; P. Rossner Jr., A. Rossnerova, M. Spatova, O. Beskid, K. Uhlirova, H. Libalova, I. Solansky, J. Topinka, R.J. Sram, Analysis of biomarkers in a Czech population exposed to heavy air pollution. Part II: chromosomal aberrations and oxidative stress, Mutagenesis 28 (2013) 97–106). With this results we would suggested possible mechanisms responsible of human adaptation to the environment, in the field of human biomonitoring that may change our view about the potential negative health impacts of the environment.

COMMENT 2 Why These people also have altered values of different markers (inflammatory for example) which can alter the results analyze the obese people?

COMMENT 3 First of all, I would analyze this possible correlation in people of normal weight

ANSWER 2-3. The SPHERE cohort is a cohort of obese subjects only. The purpose of the study was to investigate the effects of air pollutants on subjects affected by an unhealthy BMI and not to compare the possible effect among normal weight subjects and obese individuals. This choice was done as being overweight and obesity are very common conditions, with increasing prevalence worldwide, which represents a major public health issue. Moreover,  overweight/obese individuals are characterized by a higher risk of developing diseases which can be attributed to their exposure. In fact, the increased BMI could boost the response to PM exposure due to enhanced PM uptake and/or an underlying pro-oxidative state. At the molecular level, being overweight causes lipid and adipokine dysregulation, mitochondrial malfunction, induction of cellular oxidative stress. We believe that the description of the effect of PM on DNA adducts in this particular population is relevant, even if it cannot be translated to the general population.

COMMENT 4 In part 2.2. Exposure assessment is not clear how exposure is determined

ANSWER 4. We thank the reviewer for pointing out that the exposure assessment section was not clear. The methods used to assess the individual exposure of the SPHERE subjects have been described in details in Bollati et al. (doi: 10.1186/1471-2458-14-1137). However, we have expanded and rephrased the paragraph on PM exposure, to clarify the attribution of exposure. The paragraph is now as follows:

“We obtained from the Regional Environmental Protection Agency (ARPA Lombardy) recordings of daily air levels of PM10 measured from monitoring stations throughout the Lombardy region, in Italy, the day of recruitment and back to 365 days. We assigned to each participant the daily mean PM10 concentration of the monitor nearest to the Center for Obesity and Work on the day of recruitment. Individual PM10 exposure was then assigned to each subject by calculating the mean of the 30 days (short-term exposure), and of the 365 (long-term exposure) preceding enrolment. In the case of incomplete series, each missing value was imputed by using an algorithm that integrates the annual average of the incomplete series and PM10 concentrations of the nearest and more correlated monitors”.

COMMENT 5 Where do these people live? What is their daily activity? How long they stay away from home and at home (indoor pollution) etc; this impossibility of determining exposure makes it impossible to say whether or not there is a correlation between air pollution and DNA adducts.

ANSWER 5.  Study participants reside in the Lombardy Region which is situated in the Northern part of Italy, and Milan is the regional capital. Fifty percent of the subjects live in Milan and an additional 28% work in the city, even if they lived outside the city: overall 65% of subjects spent many hours a day in the city or traveling from workplace to residence. We acknowledge that a major limitation of the present study is the lack of information on the indoor PM levels, and thus we added a sentence to mention this limitation in the discussion (“A possible limitation of the present study is that we used PM10 instead of PM2.5 to estimate individual exposures because the PM10 dataset was more complete and characterized by a better spatial resolution in the study area. However, in the Lombardy Region, PM10 is mainly constituted by fine particles, and PM2.5 represents 58–94% of PM10  . In addition, unfortunately, personal exposure monitoring data are not available, due to the large study sample. Thus we could not account for indoor air pollution”).

COMMENT 6. If the calculated average values (table 2) are also correct, the exposure (except PM 10 short term, μg / m3 maximum value) can be said to be equal to or very close to the normally required value by 2008/50/CE PM10 50 µg/m3/day.

COMMENT 7. It could be a low value to see some variation; if at this value were found significant adduct DNA adducts, it would be very worrying!

ANSWER 6-7 Table 2 reports the PM10 level averaged over 30 days (short term) and over 365 days (long-term) recorded from the three monitoring stations in Milan in the Lombardy region, which is located in the Po Valley, known to be a region with high pollution levels in Italy. The EU Ambient Air Quality Directives stated a limit value of 40 µg/m3 yearly and a daily mean of 50 µg/m3 not to be exceeded on more than 35 days per year. The concentrations of particular matters in this region remain in the atmosphere for long periods and maintain much higher levels, both in terms of time and quantity, than the limits set. The Italian Po Valley situation is particularly serious and the concentrations exceeded the limits for over 50 days, with daily average values close to 50 µg/m3. (reference : https://doi.org/10.3390/earth3010013)

The mean we calculated derives from three monitors whose maximum values reach extremely high values, as reported in the table below reporting the description of the entire series of the 3 monitoring stations of Milan during the study period. Moreover, although the annual averages might have a limited variance, a great source of variability derives also from the subject-specific day of enrolment (span over 5 years). Thus, we do not fully agree with the reviewer that these levels are low or that the variation can be considered small, as seen also by the standard deviation of the mean or by the interquartile ranges.

Monitoring station

Mean

Dev std

Median

Q1

Q3

Min

Max

MILAN_PASCAL

MILAN_SENATO

MILAN_VERZIERE

44.6

45.6

46.6

29.7

32.2

31.2

36.0

36.0

37.0

23.0

23.0

25.0

59.0

60.0

59.0

2.0

1.0

7.0

213.0

203.0

228.0

COMMENT 8. Therefore, it is necessary to calculate the exposure much more precisely, treat the summer exposure differently from the winter one and divide the population into exposure groups to have reliable results.

ANSWER 8. We agree with the Reviewer as the season has an important effect both on PM levels and on the DNA adduct levels (both univariable and multivariable analysis). For this reason, we treated the variable “season” as a confounder of the model linking PM exposure to DNA adducts. A possible stratification by season would be justified by a significant interaction between PM exposure and season, which would imply an effect modifier role of the season in the PM-adduct relationship, i.e. a different relation between PM and adduct levels in the different seasons. In our study, we found no significant interaction between PM (both short- and long-term) and adducts, and thus a stratified analysis, that would reduce dramatically the sample size, doesn’t seem justified in this set of data.

We analyzed PM exposure as a continuous variable to evaluate a potential strength of association between increasing levels of PM and adducts levels. The result is an estimate of the variation in DNA adducts, associated with an increase of 1 µg/min PM concentration. A categorization of PM levels by group (e.g. quantiles, high level vs low level, etc.) will imply a loss of variability and information but lead to the same results on the linear association between PM and adduct.

Even if we would prefer to maintain the original analysis in the manuscript, we report here the plots required by the Reviewer. These plots show marginal means of the adduct, obtained from a multivariable model (adjusted for BMI, sex, diabetes, smoking habits, season, white blood cells, platelets, and apparent temperature) using PM categorization in quartiles which shows a clear decreasing trend as the quartiles of PM increase.

COMMENT 9 Figure 1 is impossible (even when zooming in on the screen) to see and understand (the methods section does not adequately describe this analysis). Is it an enlargement? What is the direction of chromatography?

ANSWER 9.

The origin (OR) of the chromatographic sheets (where the samples was applied) is located in the bottom left corner. The main chromatography directions (D1, D2) are upwards and rightwards as now indicated by arrows in the Figure 1.

COMMENT 10 Line 182-184, the authors found a positive association with air pollution exposure and DNA adducts .. showed a positive association between DNA adducts, season (in particular summer) …… in which cells?.... and apparent temperature, whereas a negative association was found with white blood cells and platelets….?

ANSWER 10

We thank the Reviewer for the note.  We changes the sentence as follows “Univariate linear regression analyses (Table 3) showed a positive association be-tween DNA adducts measured in peripheral blood leukocytes, season (in particular summer) and apparent temperature, whereas a negative association was found with white blood cells and platelets count.”

COMMENT 11 Winter PM is known to be the richest in PAH (Camatini, et al. 2010. PM10-biogenic fraction drives the seasonal variation of pro-inflammatory response in A549 cells. Environ. Toxicol., doi:10.1002/tox.20611; Perrone, et al., 2010. Seasonal variations in chemical composition and in vitro biological effects of fine PM from Milan. Chemosphere 78, 1368–1377) Consequently, if the PAHs induce DNA adducts, why is there no positive correlation between PM winter and DNA adducts, but there is with PM summer?

ANSWER 11.

We are not completely sure about the question. If we understood it properly and the question is referred to results in table 3, the analysis evaluates the changes in DNA adducts associated with the change of season, i.e. if winter is set as a reference, the adducts level is 1.28 higher in summer, but the model is not describing the association between PM and DNA adducts across the seasons,  as the model did not include PM.

The higher DNA adduct level in summer vs winter (which is significant only in the univariate model), although unexpected, might be potentially explained by other variables (not PM) that are associated with DNA adducts but distributed differently across the seasons.

COMMENT 12 The subject matter is exciting, but it is impossible to reach a conclusion with this model.

ANSWER 12

We hope that the clarifications provided above, taken as a whole, may have reassured the Reviewer of the value of the presented results.

Reviewer 3 Report

In this manuscript, Pulliero and co-authors evaluate the levels of bulky DNA adducts in white blood cells of overweight and obese individuals (MBI >= 25) recruited through the SPHERE study in Italy in order to establish effects of airborne exposure to particulate matter (<10 mm, PM10). Authors employed P32-postlabelling assay in conjunction with thin layer chromatography of DNA obtained from buffy coat layer of blood collected from 196 study subjects. Adduct levels were correlated with other variables such as PM10 exposure (short-term and long-term), smoking status, gender, and other data collected through the recruitment. The authors found inverse association between long term PM10 exposure and the levels of bulky adducts, e.g. higher level of exposure being reflected in less DNA adducts in white blood cells, explaining this hermetic effect by adipose cells sequestering lipophilic carcinogens. However, since no BMI<25 individuals were included in the study, question remains whether reported observations are actual or an artefact of analytical and statistical methods employed. Is there any internal variable/control in this study that could help answering this question? For example, it would be interesting to verify and discuss if the smoking produced similar inverse association results. Currently, this discussion is missing and data provided on the smoking association and DNA adduct levels are not conclusive.

While above were the major comments and questions, other minor comments are listed below that require further explanations and edits in the manuscript.

It is not well defined whether butanol extraction was used for DNA preparation, as the reference on the method is missing in the list of references (stated as Ref 27, while only 26 on the list). If butanol was employed, extraction should aid in removing the excess of normal nucleotides. This would affect adduct levels calculations and in the absence of a synthetic standard(s) the actual adduct levels are likely not as reported, and rather are relative levels. This could be explained better and/or referenced.

What is PM10 and its relationship to PM2.5? Authors describe PM2.5 in some detail and then switch to PM10. Is PM2.5 subset of PM10? It was not very clear from the paper; would benefit from better explaining this relationship.

Typographical errors are common across the manuscript.

Acronyms are not always explained. For example, what is “AA flare” mentioned in Discussion? In the abstract, it also could be helpful to decipher abbreviations, e.g. “PM”.

Method would benefit by explaining that the buffy coat contains white blood cells, e.g. DNA adduct levels are estimated in DNA of these cells.

The method section for exposure assessment is not well described. The following statement is somewhat confusing in the context of the paragraph. “We assign to each participant the daily mean PM10 concentration the monitor nearest to the Center for Obesity and Work”. The meaning of this sentence is not very clear. Assuming that this was the only monitor used to assign exposure, what about other monitors? Are the readings obtained from this monitor are similar to other monitors in the area?

Author Response

Reviewer 3:

COMMENT 1. In this manuscript, Pulliero and co-authors evaluate the levels of bulky DNA adducts in white blood cells of overweight and obese individuals (MBI >= 25) recruited through the SPHERE study in Italy in order to establish effects of airborne exposure to particulate matter (<10 mm, PM10). Authors employed P32-postlabelling assay in conjunction with thin layer chromatography of DNA obtained from buffy coat layer of blood collected from 196 study subjects. Adduct levels were correlated with other variables such as PM10 exposure (short-term and long-term), smoking status, gender, and other data collected through the recruitment. The authors found inverse association between long term PM10 exposure and the levels of bulky adducts, e.g. higher level of exposure being reflected in less DNA adducts in white blood cells, explaining this hermetic effect by adipose cells sequestering lipophilic carcinogens. However, since no BMI<25 individuals were included in the study, question remains whether reported observations are actual, or an artefact of analytical and statistical methods employed. Is there any internal variable/control in this study that could help answering this question? For example, it would be interesting to verify and discuss if the smoking produced similar inverse association results. Currently, this discussion is missing, and data provided on the smoking association and DNA adduct levels are not conclusive.

ANSWER 1. All diagonal radioactive zones (DRZ) detected by 32P postlabelling belongs to current smokers. Indeed, DRZ is an hallmark of cigarette smoke exposure (Izzotti A, Rossi GA, Bagnasco M, De Flora S.

(Benzo[a]pyrene diolepoxide-DNA adducts in alveolar macrophages of smokers. Carcinogenesis. 1991 Jul;12(7):1281-5.)

Accordingly, a strong relationship between eposure to cigarette smoke and presence of multiple bulky DNA adducts was detected. However, this relationship was not observed whenever al DNA adducts were considered taking into account also subjects in which the DRZ was not detected. This situation may be related to the fact that adipose tissue selectively entraps lipohilic genotoxic metaboliteds as well demonstrated for chlorophenols and diozins (Charisiadis P, Andrianou XD, van der Meer TP, den Dunnen WFA, Swaab DF, Wolffenbuttel BHR, Makris KC, van Vliet-Ostaptchouk JV. Possible Obesogenic Effects of Bisphenols Accumulation in the Human Brain. Sci Rep. 2018 May 29;8(1):8186.

However, this peculiar type of detoxification may be reversed when adipose tissue amount decrease.

COMMENT 2. It is not well defined whether butanol extraction was used for DNA preparation, as the reference on the method is missing in the list of references (stated as Ref 27, while only 26 on the list). If butanol was employed, extraction should aid in removing the excess of normal nucleotides. This would affect adduct levels calculations and in the absence of a synthetic standard(s) the actual adduct levels are likely not as reported, and rather are relative levels. This could be explained better and/or referenced.

ANSWER 2 We are grateful to the Reviewer for the suggestions. We added Reference number 27. 

COMMENT 3. What is PM10 and its relationship to PM2.5? Authors describe PM2.5 in some detail and then switch to PM10. Is PM2.5 subset of PM10? It was not very clear from the paper; would benefit from better explaining this relationship.

ANSWER 3. PM2.5 is a subset of PM10. The current literature attributes the majority of health effects linked to PM exposure to the finest components (e.g. PM2.5) and thus we mentioned the PM2.5 in the Introduction. However, in the SPHERE cohort, we used PM10 instead of PM2.5 as the air pollutant of choice because the PM10 dataset was completer and more characterized by a better spatial resolution. However, in the study area, PM10 is mainly constituted by fine particles, and PM2.5 represents 58–94% of PM10 (Bigi, A. and G. Ghermandi, Long-term trend and variability of atmospheric PM10 concentration in the Po Valley. Atmospheric Chemistry and Physics, 2014. 14(10): p. 4895-4907).

COMMENT 4. Acronyms are not always explained. For example, what is “AA flare” mentioned in Discussion? In the abstract, it also could be helpful to decipher abbreviations, e.g. “PM”.

ANSWER 4. We are grateful with the Reviewer for the suggestions. We specified the abbreviations as suggested. The acronym (AAs) is aromatic amines and particular matter (PM).

COMMENT 5. Method would benefit by explaining that the buffy coat contains white blood cells, e.g. DNA adduct levels are estimated in DNA of these cells.

ANSWER 5. We thank the Reviewer for the note. Buffy-coat derived white blood cells collected at baseline.

We included this information in the method section as follows:

“Genomic DNA was extracted from the buffy coat fraction (including lymphocytes, monocytes, granulocytes, and platelets) using the Wizard Genomic DNA Purification Kit (Promega; Madison, WI, USA) according to the manufacturer’s instructions and used to perform DNA adduct quantification.”

COMMENT 6. The method section for exposure assessment is not well described. The following statement is somewhat confusing in the context of the paragraph. “We assign to each participant the daily mean PM10 concentration the monitor nearest to the Center for Obesity and Work”. The meaning of this sentence is not very clear. Assuming that this was the only monitor used to assign exposure, what about other monitors? Are the readings obtained from this monitor are similar to other monitors in the area?

ANSWER 6. We thank the Reviewer for pointing out the clarity of this part of the manuscript. The methods used to assess the individual exposure of the SPHERE subjects have been described in detail in Bollati et al. (doi: 10.1186/1471-2458-14-1137). However, we have expanded and rephrased the paragraph on PM exposure, to clarify the attribution of exposure. The paragraph is now as follows:

“We obtained from the Regional Environmental Protection Agency (ARPA Lombardy) recordings of daily air levels of PM10 measured from monitoring stations throughout the Lombardy region, in Italy, the day of recruitment and back to 365 days. We assigned to each participant the daily mean PM10 concentration of the monitor nearest to the Center for Obesity and Work on the day of recruitment. Individual PM10 exposure was then assigned to each subject by calculating the mean of the 30 days (short-term exposure), and of the 365 (long-term exposure) preceding enrolment. In the case of incomplete series, each missing value was imputed by using an algorithm that integrates the annual average of the incomplete series and PM10 concentrations of the nearest and more correlated monitors”.

Round 2

Reviewer 2 Report

see attach 

PPP

Author Response

Dear Editors,

We would like to thank you for considering the manuscript entitled “The relationship between exposure to airborne particulate and DNA adducts in blood cells in an urban population of subjects with an unhealthy body mass index”

” by Pulliero A. et al., and for sharing the Reviewers’ comments that certainly helped in improving the quality of the manuscript (ID: ijerph-1668622). We appreciated the Reviewers’ comments, and we revised the manuscript accordingly. Please find enclosed to the submission of the revised version of the manuscript the point-by point reply to the Reviewers’ comments. For clarity’s sake, changes in the revised MS are wrote in blue color.

We hope that this revised version of our MS will be now suitable for publication in the IJERPH.

Accordingly, we prepared a revised version of the manuscript acknowledging Referees’ and Editor’s comments as below specified:

COMMENT 1. The authors conclude that… Overall, the 25 findings of the linear univariate and multiple linear regression showed an inverse association between long-term PM10 exposure and adduct levels.  The unexpected results, in my opinion, maybe due to an incorrect experimental model.

Rev response:  The authors cite 2 papers about 10 years ago that found similar results in healthy individuals, NOT OVERWEIGHT.

The Czech authors correlate the concentration of B[a]P and benzene and DNA adducts not PM10 and DNA adducts It is the principal response to DNA adducts are B[a]P and benzene, present mainly in PM2, 5 slightly in PM10 (Perrone, et al 2010. Seasonal variations in chemical composition and in vitro biological effects of fine PM from Milan. Chemosphere 78, 1368 1377.; Camatini et 2010 https://doi.org/10.1002/tox.20611; Gualtieri et al., https://doi.org/10.1016/j.toxlet.2009.03.003) I must object that the Czech authors correlate the concentration of B[a]P and benzene and DNA adducts, not always finding a negative correlation. Pulliero et al. are aware of the B[a]P most responsible for the adoptions to which their overweight population has been exposed ?

Did they evaluate the levels of other parameters in the blood, as Czech authors, to support their unusual explanation? Mutagenesis vol. 28 no. 1 pp. 89 95, 2013 Advance Access publication 9 October 201 Multivariate analyses conducted among subjects from Ostrava and Prague separately during all sampling periods revealed that exposure to B[a]P and PM2.5 significantly increased levels of B[a]P-like DNA adducts in the Ostrava subjects, but not in subjects ... from Prague.

This unexpected observation indicates changes in lifestyle of study subjects for the reasons that are not clear. Theoretically, the differences in concentrations of vitamins in blood plasma may have affected bulky DNA adduct levels.

A recent review indicates that vitamin E is inversely associated with DNA adduct levels, whereas vitamins A and C are not independent predictors of DNA adducts.

ANSWER 1 BIS. Conclusions have been expanded now explaining that the unexpected inverse association between long-term PM10 exposure and adduct levels may be related to (a) trapping of lipophilic genotoxic into adipose tissue [34, 35] (b) adaptive mechanisms triggered by low level environmental exposure. The high body mass index makes low an exposure that person having a normal or low BMI would higher, as an example occurring in children.

The influence of body fat on toxicodynamic of environmental pollutants have never been explored insofar. Trapping of lipophilic pollutants inside fat tissue does not represent a detoxification; indeed, these genotoxicants may be released again from adipose tissue into the blood stream especially when decrease of fat tissue occurs. These toxicodynamic mechanisms are well known for lipophilic carcinogens such as chlorophenols [35]. However, insofar no data has been collected dealing PM.

Exposure of our subjects was to a complex mixture of airborne pollutants including a small contribution from B(a)P. This is the reason because the 32P post-labelling method was used for DNA adduct analysis. Indeed, this is the only method able to (a) evaluate all together multiple adducts formed by exposure to mixture; (b) detect adducts of unknown molecular structure. These statements have been added in the text (Discussion, lines 291-301).

COMMENT 2 Why These people also have altered values of different markers (inflammatory for example) which can alter the results analyze the obese people?

ANSWER 2 BIS. It is well established that obesity cause systemic inflammation. A sentence reporting the point highlighted by the reviewer has been added in Discussion (line 290) and supported by a newly added reference [35].

COMMENT 3 First of all, I would analyze this possible correlation in people of normal weight

ANSWER 2-3. The SPHERE cohort is a cohort of obese subjects only. The purpose of the study was to investigate the effects of air pollutants on subjects affected by an unhealthy BMI and not to compare the possible effect among normal weight subjects and obese individuals. This choice was done as being overweight and obesity are very common conditions, with increasing prevalence worldwide, which represents a major public health issue. Moreover, overweight/obese individuals are characterized by a higher risk of developing diseases which can be attributed to their exposure. In fact, the increased BMI could boost the response to PM exposure due to enhanced PM uptake and/or an underlying pro-oxidative state. At the molecular level, being overweight causes lipid and adipokine dysregulation, mitochondrial malfunction, induction of cellular oxidative stress. We believe that the description of the effect of PM on DNA adducts in this particular population is relevant, even if it cannot be translated to the general population.

Rev response I am well aware of the problem of people with NCDs exposed to air pollution, but I believe that it is not the ideal model to analyze the correlation between air pollution and DNA adduct if not in comparison to a healthy population. Any data could be altered by an inflammatory condition, as state in the answer… At the molecular level, being overweight causes lipid and adipokine dysregulation, mitochondrial malfunction, induction of cellular oxidative stress.

ANSWER 2-3 BIS. We fully agree with the reviewer that overweight subjects exhibit a range of changes related to their chronic level of low-grade inflammation. For this reason, we used the BMI as a covariate of adjustment in all the statistical models. However, what we observe in this work should not be considered a concept that can be extended to the general population. In fact, from the title of the work, we declare that the study is conducted on this specific population of subjects (Title: “The relationship between exposure to airborne particulate and DNA adducts in blood cells in an urban population of subjects with an unhealthy body mass index”).

The current literature supports the positive association between particulate levels and adduct quantity in populations with a normal BMI (16,17,24). Therefore, the aim of this study was to study the subjects with high BMI.

The relationship between NCD risk, DNA adduct formation has been already explored and established in healthy population. Herein presented results, indicate that obesity is a confounding factor able to influence the relationship between DNA adduct formation and exposure to environmental pollutants. This statement has been added in Conclusions (lines 327-335).

COMMENT 4 In part 2.2. Exposure assessment is not clear how exposure is determined

ANSWER 4. We thank the reviewer for pointing out that the exposure assessment section was not clear. The methods used to assess the individual exposure of the SPHERE subjects have been described in detail in Bollati et al. (doi: 10.1186/1471-2458-14-1137). However, we have expanded and rephrased the paragraph on PM exposure, to clarify the attribution of exposure. The paragraph is now as follows:

We obtained from the Regional Protection Agency (ARPA Lombardy) recordings of daily air levels of PM10 measured from monitoring stations throughout the Lombardy region, in Italy, the day of recruitment and back to 365 days. We assigned to each participant the daily mean PM10 concentration of the monitor nearest to the Center for Obesity and Work on the day of recruitment. Individual PM10 exposure was then assigned to each subject by calculating the mean of the 30 days (short-term exposure), and of the 365 (long-term exposure) preceding enrolment. In the case of incomplete series, each missing value was imputed by using an algorithm that integrates the annual average of the incomplete series and PM10 concentrations of the nearest and more correlated minitors.

Rev response the answer is acceptable but I insist: a) PM10 is measured and not PM2.5 or BaP; b) it is never clear how much people have been exposed; this approach could have been acceptable 10 years ago now, not anymore.

ANSWER 4 BIS. As also mentioned in the manuscript, we used PM10 instead of PM2.5 to estimate individual exposures because the PM10 dataset was more complete and characterized by a better spatial resolution in the study area, at least in the years of the study. The percentages indicated are derived from the cited papers. However, if we select the series of PM2.5 and PM10 of the  “Milano Pascal” monitoring station, chosen as it is the only one in Milan with both PM2.5 and PM10 available on the majority of the days of the study period (n=923), and estimate the Pearson correlation coefficient we obtain a very high correlation between the two series (rho=0.95532; p-value<.0001). Moreover, calculating the main ratio between PM10 and PM2.5, over the study period PM2.5 is the 69% of PM10. The following graph reports the correlation between the two series. Finally, BaP series are largely incomplete and would generate too many missing values.

COMMENT 5 Where do these people live? What is their daily activity? How long they stay away from home and at home (indoor pollution) etc; this impossibility of determining exposure makes it impossible to say whether or not there is a correlation between air pollution and DNA adducts.

ANSWER 5. Study participants reside in the Lombardy Region which is situated in the Northern part of Italy, and Milan is the regional capital. Fifty percent of the subjects live in Milan and an additional 28% work in the city, even if they lived outside the city: overall 65% of subjects spent many hours a day in the city or traveling from workplace to residence. We acknowledge that a major limitation of the present study is the lack of information on the indoor PM levels, and thus we added a sentence to mention this limitation in the discussion (A possible limitation of the present study is that we used PM10 instead of PM 2.5 to estimate

individual exposures because the PM10 dataset was more complete and characterized by a better spatial resolution in the study area. However, in the Lombardy Region, PM10 is mainly constituted by fine particles, and PM2.5 represents 58 94%??????? of PM10 . In addition, unfortunately, personal exposure monitoring.

Rev response The answer is acceptable but I insist

  1. PM10 is measured and not PM2.5 or BaP; b) it is never clear how much people have been exposed; this approach could have been acceptable 10 years ago now, not anymore. The share of PM2.5 contained in PM10 can vary from 50-60% in winter to 40% in summer; this share also includes UFPs (MILAN) But according to EPA regulation, the efficiency of the sampling system MUST contain at least 50% of particles having the Dae of = 10 um (PM10).

ANSWER 5 BIS. This Question Is the same that Question 4. As also mentioned in the manuscript, we used PM10 instead of PM2.5 to estimate individual exposures because the PM10 dataset was completer and more characterized by a better spatial resolution in the study area, at least in the years of the study. The percentages indicated are derived from the cited papers. However, if we select the series of PM2.5 and PM10 of the  “Milano Pascal” monitoring station, chosen as it is the only one in Milan with both PM2.5 and PM10 available on the majority of the days of the study period (n=923), and estimate the Pearson correlation coefficient we obtain a very high correlation between the two series (rho=0.95532; p-value<.0001). Moreover, calculating the main ratio between PM10 and PM2.5, over the study period PM2.5 is the 69% of PM10. The following graph reports the correlation between the two series. Finally, BaP series are largely incomplete and would generate too many missing values.

COMMENT 6. If the calculated average values (table 2) are also correct, the exposure (except PM 10 short term, g / m3 maximum value) can be said to be equal to or very close to the normally required value by 2008/50/CE PM10 50 µg/m3 /day.

COMMENT 7. It could be a low value to see some variation; if at this value were found significant adduct DNA adducts, it would be very worrying!

ANSWER 6-7 Table 2 reports the PM10 level averaged over 30 days (short term) and over 365 days (long term) recorded from the three monitoring stations in Milan in the Lombardy region, which is located in the Po Valley, known to be a region with high pollution levels in Italy. The EU Ambient Air Quality Directives stated a limit value of 40 µg/m3 yearly and a daily mean of 50 µg/m3 not to be exceeded on more than 35 days per year. The concentrations of particular matters in this region remain in the atmosphere for long periods and maintain much higher levels, both in terms of time and quantity, than the limits set. The Italian Po Valley situation is particularly serious and the concentrations exceeded the limits for over 50 days, with daily average values close to 50 µg/m3. (reference: https://doi.org/10.3390/earth3010013) The mean we calculated derives from three monitors whose maximum values reach extremely high values, as reported in the table below reporting the description of the entire series of the 3 monitoring stations of Milan during the study period. Moreover, although the annual averages might have a limited variance, a great source of variability derives also from the subject-specific day of enrolment (span over 5 years). Thus, we do not fully agree with the reviewer that these levels are low or that the variation can be considered small, as seen also by the standard deviation of the mean or by the interquartile ranges.

Monitoring station

Mean

Dev std

Median

Q1

Q3

Min

Max

MILAN_PASCAL

MILAN_SENATO

MILAN_VERZIERE

44.6

45.6

46.6

29.7

32.2

31.2

36.0

36.0

37.0

23.0

23.0

25.0

59.0

60.0

59.0

2.0

1.0

7.0

213.0

203.0

228.0

Rev response I know the situation in the Po Valley very well; the averaged values, even with high SD, were "quite good", taking into account that the values are often higher (November-March). But, I insist how you can tell what level of PM10 (not PM2.5) individuals have been exposed by averaging on 3 Monitoring stations located in different Milan areas and draw conclusions.

See Answer 5  BIS

COMMENT 8. Therefore, it is necessary to calculate the exposure much more precisely, treat the summer exposure differently from the winter one and divide the population into exposure groups to have reliable results.

ANSWER 8. We agree with the Reviewer as the season has an important effect both on PM levels and on the DNA adduct levels (both univariable and multivariable analysis). For this reason, we treated the variable “season” as a confounder of the model linking PM exposure to DNA adducts. A possible stratification by season would be justified by a significant interaction between PM exposure and season, which would imply an effect modifier role of the season in the PM-adduct relationship, i.e. a different relation between PM and adduct levels in the different seasons. In our study, we found no significant interaction between PM (both short- and long-term) and adducts, and thus a stratified analysis, that would reduce dramatically the sample size, doesn’t seem justified in this set of data.

We analyzed PM exposure as a continuous variable to evaluate a potential strength of association between increasing levels of PM and adducts levels. The result is an estimate of the variation in DNA adducts, associated with an increase of 1 µg/min PM concentration. A categorization of PM levels by group (e.g. quantiles, high level vs low level, etc.) will imply a loss of variability and information but lead to the same results on the linear association between PM and adduct.

Even if we would prefer to maintain the original analysis in the manuscript, we report here the plots required by the Reviewer. These plots show marginal means of the adduct, obtained from a multivariable model (adjusted for BMI, sex, diabetes, smoking habits, season, white blood cells, platelets, and apparent temperature) using PM categorization in quartiles which shows a clear decreasing trend as the quartiles of PM increase.

Rev response I insist how you can tell what level of PM10 (not PM2.5) individuals have been exposed by averaging on 3 Monitoring stations located in different Milanese areas and draw conclusions?

See Answer 5. BIS

COMMENT 12 The subject matter is exciting, but it is impossible to reach a conclusion with this model. ANSWER 12 We hope that the clarifications provided above, taken as a whole, may have reassured the

Rev response Many points have been clarified and improved in the paper, but the experimental model remains, now in the 2022, inadequate to analyze the associations between PM10 (better would be PM2.5) and DNA adduct and draw conclusions. a) the population itself is a low-inflammatory population

  1. b) the degree of exposure to PM10 is unclear and and not determined accurately; this means that the level of exposure and for how long is not absolutely certain
  2. c) the PAHs responsible for DNA adduct are more present in PM2.5 (not in PM10), and it is incorrect to indicate that PM2.5 represents 58 94% ??????? of PM10.

ANSWER 12 BIS. The genotoxicity of PM10 can vary spatially and temporally, which was demonstrated in nonsmoker residents of Mexico City, and that presented significantly higher levels of DNA adducts in the dry season compared to the rainy season, because PM10 concentrations decreased during the wet station [12]. Similar results have been obtained in cell models since PM10 collected in winter induce higher levels of PAH-DNA adducts in comparison to samples collected in summer in hepatocellular carcinoma cells (HepG2) [13,14]. On the other hand, PM10 from an industrial zone induced twice the formation of DNA adducts compared to PM10 obtained from a residential area of the same city [15]. These results show that measurement of DNA adduct levels can reflect the average exposure to PM and suggest that DNA damage is closely related to the intensity of air pollution and likely with PAH concentrations [16].

Levels of PAH tend to be higher in PM of smaller aerodynamic diameter (PM2.5), however, studies show that PM with a larger diameter (PM10) also induce the formation of adducts. Lung epithelial A549 and BEAS-2B cells showed DNA PAH-Adducts after exposure to PM10 and PM2.5 associated with PAH bioactivation, which is reflected in increased expression of the CYP1A1 and CYP1B1 gene and protein activity after 24 h of exposure to PM10 and 24, 48 and 72 h of exposure to PM2.5 [17,18].

Reviewer 3 Report

The reviewer thank authors for addressing his/her concerns regarding the manuscript. These are potentially very important observations that warrant further research in various groups of people with unhealthy as well as healthy BMI. Also, the smoking angle and BMI is interesting as well.

There are some minor typos remaining, but other than that my questions have been answered.

Author Response

We wish to thank the Reviewer for the kind and positive comments. We have revised the manuscript, considering the Reviewers’ concerns. 
